# The Effect of Temperature Cycling on the Magnetic Degradation and Microstructure of a Zn-Coated NdFeB Magnet

**Hu-Lin Wu [1], Zhi-Mei Long [2], Kai-Qiang Song [1], Chao-Qun Li [2], Da-Long Cong [1], Bin Shao [2,*], Xiao-Wei Liu [2] and Yi-Long Ma [2]**

1   Southwest Institute of Technology and Engineering, Chongqing 400039, China; wuhulin59@163.com (H.-L.W.); scut_song@163.com (K.-Q.S.); congdl09@163.com (D.-L.C.)
2   School of Metallurgy and Material Engineering, Chongqing University of Science and Technology, Chongqing 401331, China; 2021202024@cqust.edu.cn (Z.-M.L.); 2020202022@cqust.edu.cn (C.-Q.L.); lxwicq@163.com (X.-W.L.); yilongma@163.com (Y.-L.M.)
*   Correspondence: shaobin19811107@163.com

**Abstract:** Temperature cycling tests in various temperature ranges were carried out to investigate the magnetic degradation of the Zn-coated NdFeB magnet. The losses of the surface magnetic field and magnetic flux were well fitted by using an index model. Compared with the lower limit temperature, the upper limit temperature had more obvious effect on the magnetic degradation. Once the upper limit temperature exceeded $\geq 160\,^{\circ}\mathrm{C}$, the magnetic degradation mainly occurred during the first cycle, which was different from the gradual decline with an increase in cycle number at a temperature of $\leq 140\,^{\circ}\mathrm{C}$. Moreover, the temperature cycling with a maximum upper limit temperature of $180\,^{\circ}\mathrm{C}$ led to a loss of the remanence intensity, while the coercivity remained stable. Microstructure and element distribution analysis revealed that the oxidation of the Zn coating layer during the temperature cycling causes its cracking and an insertion of the oxygen element into the NdFeB substrate. The Nd-, Pr-rich phase at grain boundaries provided diffusion channels for oxygen elements, leading to a surface oxidation of NdFeB grains.

**Keywords:** NdFeB magnet; Zn coating; thermal cycling; magnetic degradation





## 1. Introduction

Sintered bulk NdFeB based magnets with excellent magnetic performance have been extensively utilized for traction motors in hybrid vehicles and electric vehicles [1,2]. For most driving motors, NdFeB magnets need to be exposed to an environment where the temperature is higher than room temperature for long periods due to the heat generated by eddy current losses [3,4]. The temperatures in the interior and engine room can be raised to the range 85–200 °C [5,6], whereas NdFeB magnets cannot work in these high temperature operating environments owing to their low curie temperature (about 315 °C) and poor thermal stability. The partial substitution of Nd with Dy, Pr, Td, and other elements can increase both Curie temperature and anisotropy field of NdFeB-based magnets [7–9]. Nevertheless, it is also important for the design engineer to understand what irreversible losses are incur ed at elevated temperatures [10].

Continuous eddy currents maintain the operating temperature close to the critical temperature of the magnets, which causes a time-dependent degradation through the magnetic viscosity [11]. Many studies have focused on investigating the effects of elevated temperature air exposure on the decay law of its magnetic decay law of the magnetics [12–15] and the evolution of their microstructures [16–20] during long-term aging. Li et al. [16] found that two external-oxide layers composed of $Fe_2O_3$ and $Fe_3O_4$ and an internal oxidation zone (IOZ) were generated on the surface of the bare NdFeB magnet, but these had no protective effect for the internal oxidation. The formation of IOZ is caused by the dissociation of the $Nd_2Fe_{14}B$ phase to form oxides of neodymium and boron in a matrix

of un-oxidized iron [16,19], and the columnar $\alpha$-Fe grains oriented at right-angles to the specimen surface were believed to provide short-circuit diffusion paths for the inward transport of oxidation [16]. A logarithmic law can be well used to fit a time-dependent flux loss for the bare NdFeB magnet [20]. On the other hand, motors in actual operation do not have a constant and continuous output but repeat start–stop cycles, leading the temperature in the motor to continuously switch between an ambient and working temperatures [21]. Fujiwara et al. [22] showed a 4% remanence loss of the bare NdFeB thick film occurred during the 1st cycling between the room temperature and 70 °C, after which the remanence remains essentially unchanged.

The bare NdFeB magnets exposed at an elevated temperature above 700 °C are easier to get the information of microstructural changes, but the commercial NdFeB magnets are usually coated by a protective layer to improve their thermal ability [23]. Nababan et al. [17] proved that a Ni/Cu/Ni coating layer had a significant effect in reducing the overall oxidation kinetic rate by ten times, and that the rate-limiting steps corresponded to the complexity of the microstructure of the oxidation layers depended on the elevated temperature. Huger and Gerling [24] showed that the flux losses of the NdFeB magnet with different coating layers caused by thermal cycling in −30–150 °C range was also a logarithmic type and it was speculated that the influence of the thermal aftereffects during the 1st cycle was separated from the aging mechanism.

There are still few studies on the temperature cycling of the Zn-coated NdFeB magnets. In most case, only one upper limit temperature is chosen for the temperature cycling test, and the microstructure information of the coating surface and NdFeB substrate during temperature cycling below 200 °C is also very limited. In this study, magnetic degradation of the commercial Zn-coated NdFeB magnet is investigated through a temperature cycling test in different temperature ranges. A power index model is formed to describe the relation of flux and surface magnetic field losses with cycle number. Finally, the microstructure characterization is carried out to understand the oxidation behavior of the Zn-coated NdFeB magnet during temperature cycling.

## 2. Experiment and Characterization

### 2.1. Materials

Sintered bulk NdFeB magnets (40UH) with a Zn coating layer that were previously used in motor stators were obtained from Hubei Permanent Magnet Technology Co. Ltd. (China). The sample dimensions were 28 × 24 × 4.5 mm$^3$, and compositions of the sample is presented in Table 1. The residual magnetization strength ($B_r$) of the original sample is 1.69 kGs, and the coercive force ($H_{cj}$) of that is 22.86 kOe. The magnetic flux ($\Phi$) is 20.38 mWb, and the surface magnetic field ($B$) is 191 mT. Electro-galvanizing was applied to form the Zn coating layer with a thickness of about 3 μm on the surface of the bare NdFeB magnet.

**Table 1.** Chemical compositions of the Zn-coated NdFeB sample used in this study (at.%).

| Sample | Fe | Nd | Pr | Dy | Al | B |
|---|---|---|---|---|---|---|
| NdFeB | 83.95 | 10.59 | 2.44 | 1.27 | 0.86 | 0.89 |

### 2.2. High-Temperature Test

A high-temperature test was carried out in a thermostatic furnace. Each sample was placed in an aluminum sample box with a certain distance to eliminate the magnetic effect. When the thermostatic furnace reached the set temperature, the sample was placed in the furnace hot zone at the required temperature (80, 120, 150, 180, 220, 250, and 300 °C) for 1 h. Finally, the samples were cooled to room temperature in natural air.

### 2.3. Temperature Cycling Test

For temperature cycling tests, a high-temperature incubator and a low-temperature incubator were used to reduce the influence of the heating and cooling times, respectively. All samples were firstly placed in the low-temperature incubator and kept at the set temperature for 1 h. Then, the samples were transferred to the high-temperature incubator within 1 min and also kept at the set temperature for 1 h. The entire process is defined as one cycle. The lower temperatures were set as −50 °C and room temperature (R.T.), while the upper limit temperatures were set as 120, 140, 160, and 180 °C, respectively. The samples after 10, 20, 40, 60, 80, 100, and 125 cycles were taken for further characterization.

### 2.4. Characterization

The microstructures of the original and recycled Zn-coated NdFeB magnets were visualized by a field emission scanning electron microscopy (FE-SEM, JSM-7800F, Japan Electronics JEOL, Tokyo, Japan). The elemental distribution and semi-quantitative chemical composition analyses were characterized by energy dispersive X-ray spectrometry (EDS, Oxford Instruments, Abingdon, UK) associated with SEM using an X-MaxN 80 T detector (Oxford Instruments, Abingdon, UK). All samples were demagnetized for SEM observation. The magnetic flux ($\Phi$) of the magnet was measured using a Maxwell meter (YC-820, Hunan Permanent Magnet Measurement and Control Technology Co., Ltd, Hunan, China) with Helmholtz coils (MM-150D, Hunan Permanent Magnet Measurement and Control Technology Co., Ltd, Hunan, China), and the surface magnetic field ($B$) at the center position of the magnet was measured with a Teslameter (LZ-610H, Hunan Permanent Magnet Measurement and Control Technology Co., Ltd, Hunan, China). Demagnetization curves of the bulk samples was tested using a magnetograph (AMT-4, Mianyang National High-tech Zone Bipolar Electronics Co., Ltd, Sichuan, China) and the relevant magnetic performance parameters were also obtained. The remanence ($B_r$) is the magnetic induction intensity retained in the material when the magnetic field is reduced to zero after magnetization, which is represented the ability of magnetic materials to resist demagnetization. The coercivity ($H_{cj}$) is the reverse magnetic field that reduces the magnetic induction intensity to zero. The maximum energy product (($BH)_{max}$) is the maximum value of $BH$ obtained in the demagnetization curve, which is the maximum amount of useful work that can be performed by the magnet. The squareness ($H_k/H_{cj}$) is defined by the ratio of the reverse field required to reduce $B$ by 10% from the remanence to $H_{cj}$, which is a measure of how square the curve.

The average values calculated from three samples were used for each data point. Moreover, the losses ($Y$) of the $\Phi$ or $B$ was further defined as $Y = \frac{Y' - Y_0}{Y_0} \times 100\%$, where $Y'$ and $Y_0$ represent the $\Phi$ or $B$ of the recycled and original samples, respectively.

## 3. Results and Discussion

### 3.1. Effect of Temperature Cycling on Magnetic Properties

A high-temperature test is firstly performed to contrast with the results of the temperature cycling experiments. Figure 1 shows the values of $\Phi$, $B$ and their losses from R.T. to 300 °C. As the upper limit temperature increases from 150 °C to 180 °C, the loss of $B$ increases from 1.78% to 19.02%, while the attenuation rate of $\Phi$ increases from 1.52% to 3.93% (Table S1). The losses of both $\Phi$ and $B$ increase rapidly from 180 °C.

The upper limit temperature is fixed at 180 °C for the temperature cycling test, and the effect of the lower limit temperature on the $\Phi$ and $B$ of the Zn-coated NdFeB magnet is firstly investigated. The results (Figure S1 and Tables S2 and S3) show that the loss difference of $\Phi$ and that of $B$ between −50–180 °C and R.T.–180 °C are only 0.46% and 0.29% after the 125th cycle, respectively. This demonstrates that the lower limit temperature has a very limited influence on the magnetic losses during the temperature cycling. In the following experiments, the lower limit temperature was fixed at R.T. The values of $\Phi$ and $B$ and their losses at different upper limit temperatures are shown in Figure 2. At 120, 140, 160, and 180 °C, the losses of $\Phi$ are 0.23, 0.76, 0.97, 19.13, and 27.39% after ten cycles,

and those of the $B$ are 0.38, 1.88, 2.10, 39.70, and 43.11% (Table S4). After the 125 cycles, the losses of $\Phi$ increase to 1.29, 1.58, 1.78, 24.10, and 33.74%, while those of $B$ increased to 4.78, 5.02, 14.12, 46.18, and 49.44% (Table S4). At the upper limit temperature of $\geq$160 °C, the losses of $\Phi$ and $B$ mainly occur before the 10th cycle, but increase gradually with the cycle number at the upper limit temperature $\leq$ 140 °C. Furthermore, $B$ is found to be more sensitive to the magnetic degradation than $\Phi$.

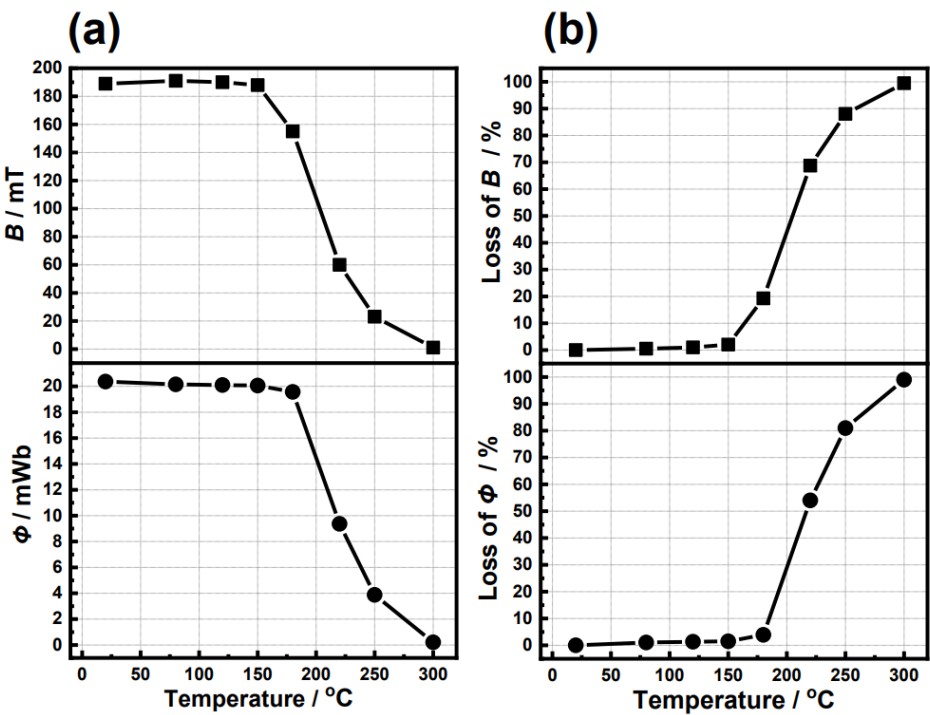

**Figure 1.** Temperature dependence of (**a**) $\Phi$, (**b**) $B$ and their losses of the samples from R.T. to 300 °C.

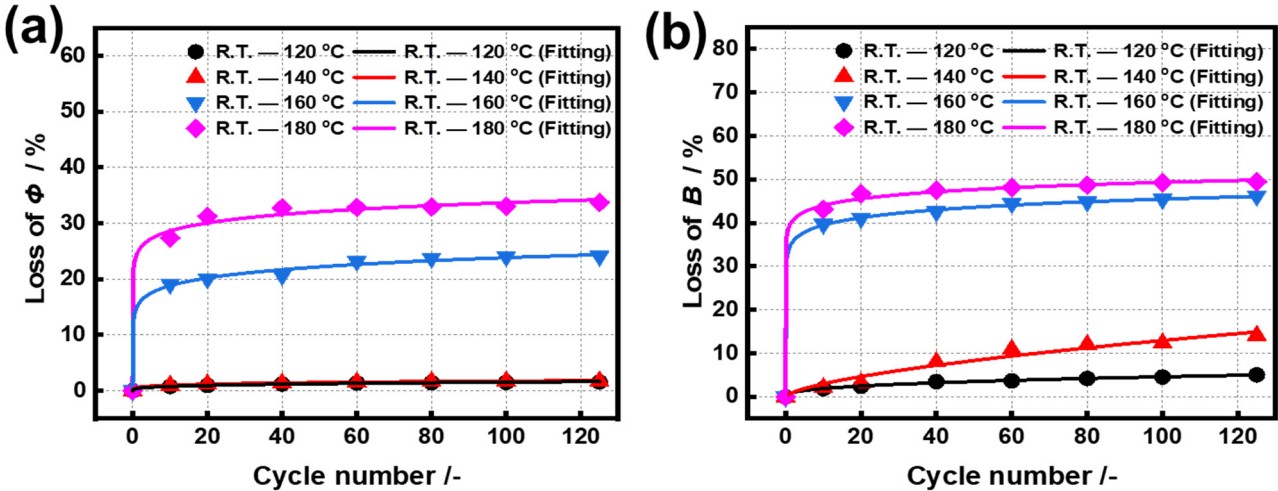

**Figure 2.** Losses of (**a**) $\Phi$ and (**b**) $B$ of the samples cycled in ranges of R.T. to 120, 140, 160, and 180 °C, respectively. The lines in figures are the fitting curves with a formula $Y = A \times (X - X_c)^P$.

The formula $Y = A \times (X - X_c)^P$ is applied to fit the experimental data, where $Y$ represents the losses of $\Phi$ or $B$, and $X$ is the cycle number, and $A$, $X_c$ and $P$ are the fitting parameters. The fitting curves are shown in Figure 3b,d, and the values of the fitted parameters are given in Table 2. All the curves exhibit good fitting results with $R^2 \geq 0.95$, and the values of $X_c$ are mostly all 0.00. As the upper limit temperatures increase from

140 °C to 160 °C, the values of $A$ increase from 0.57 to 14.85, and the values of $P$ decrease from 0.24 to 0.01, meaning that the magnetic degradation mainly occur in the early stages of the temperature cycling. According to the above fitting results, the values of losses of $\Phi$ and $B$ under 1st, 500th, and 1000th cycles are further simulated as shown in Table 3. At the upper limit temperatures of 120 °C and 140 °C, the losses of $\Phi$ are only 0.39% and 0.57% after the 1st cycle and are 3.06% and 3.10% after the 1000th cycle. Once the upper limit temperature reaches to 160 °C, the loss of $\Phi$ immediately rises to 14.86% after the 1st cycle, and then gradually increases to 30.22% after 1000th cycle. The previous study reports a 4% remanent loss after the 1st cycle between R.T. and 70 °C by using a bare NdFeB film as a sample [22]. Obviously, the Zn coating layer can effectively improve the thermal stability of the NdFeB magnet during the temperature cycling, but 140 °C seems to be the upper limit.

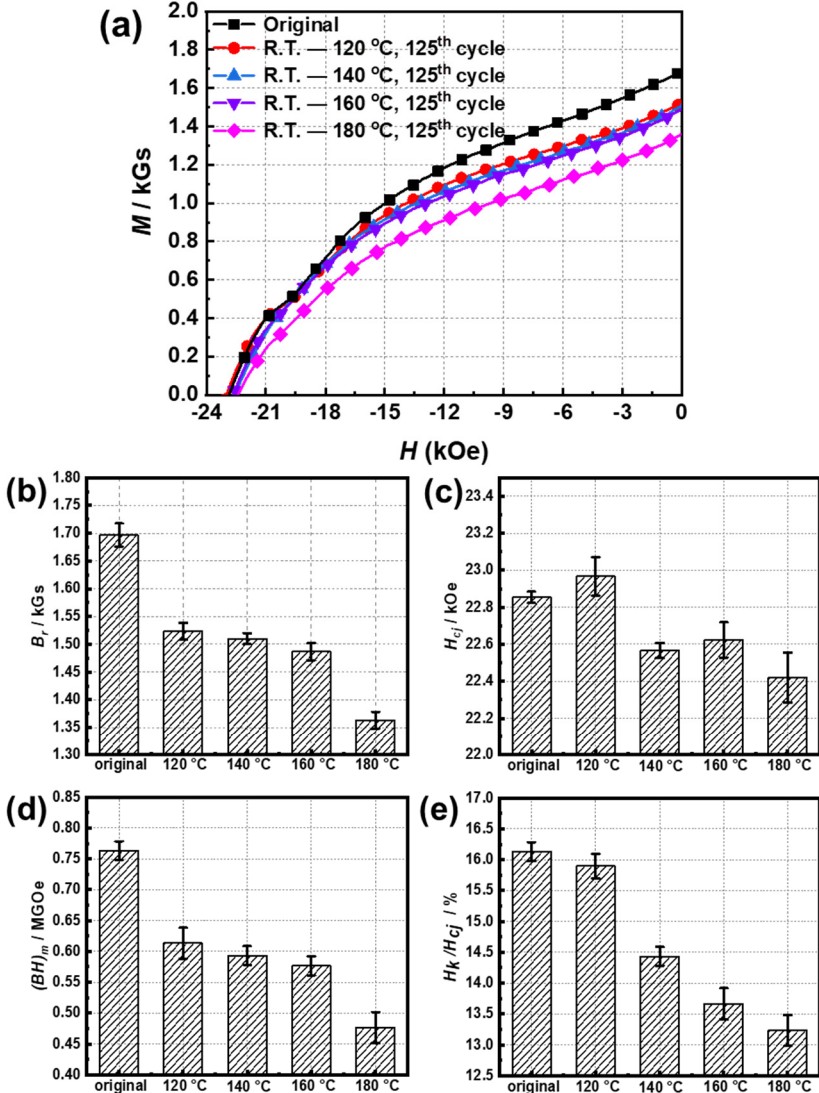

**Figure 3.** (**a**) Demagnetization curves, (**b**) $B_r$, (**c**) $H_{cj}$, (**d**) $(BH)_{max}$, and (**e**) $H_k/H_{cj}$ of the samples after the 125th cycle at different upper limit temperatures.

Figure 3a shows the demagnetization curves of the demagnetized samples cycled in the temperature ranges of R.T. to 120, 140,160, and 180 °C after the 125th cycle, respectively. It finds that the values of coercivity ($H_{cj}$) remain in the range 22.42–22.97 kOe, independent of the upper limit temperature. For the bare (NdDy)FeB magnet, its $H_{cj}$ decreases significantly even at 60 °C for 2 h [13]. On the other hand, the increase of the upper limit temperature remarkably leads to a significant reduction in the residual magnetization ($B_r$), squareness

$(H_k/H_{cj})$ of the magnet and the maximum magnetic energy $((BH)_{max})$. This result can explain the decrease of $\Phi$ and $B$ with the increase in the upper limit temperature, because both of which are positively proportional to $B_r$.

**Table 2.** Values of fitting parameters and their standard errors (*S.E.*) in the different temperature ranges. The fitting formula is given by $Y = A \times (X - X_c)^P$.

| $Y$ | Temperature Range | Fitting Parameter | | | | | | |
|---|---|---|---|---|---|---|---|---|
| | | $X_c$ | $S.E._{\cdot(X_c)}$ | $A$ | $S.E._{\cdot(A)}$ | $P$ | $S.E._{\cdot(P)}$ | $R^2$ |
| Loss of $\Phi$ | R.T.–120 °C | 0.00 | 6.46 | 0.39 | 0.12 | 0.30 | 0.06 | 0.99 |
| | R.T.–140 °C | 0.00 | 4.86 | 0.57 | 0.11 | 0.24 | 0.04 | 0.99 |
| | R.T.–160 °C | 0.00 | 11.01 | 14.85 | 2.77 | 0.01 | 0.04 | 0.99 |
| | R.T.–180 °C | 0.00 | 18.18 | 24.33 | 5.22 | 0.07 | 0.05 | 0.99 |
| Loss of $B$ | R.T.–120 °C | 0.00 | 4.72 | 0.78 | 0.21 | 0.39 | 0.06 | 0.99 |
| | R.T.–140 °C | 0.13 | 1.25 | 0.70 | 0.29 | 0.63 | 0.09 | 0.95 |
| | R.T.–160 °C | 0.00 | 4.58 | 34.29 | 1.61 | 0.06 | 0.01 | 0.99 |
| | R.T.–180 °C | 0.00 | 12.85 | 39.30 | 4.19 | 0.05 | 0.02 | 0.99 |

**Table 3.** Simulation results of the losses of $\Phi$ and $B$ in different temperature ranges after the 1st, 500th, and 1000th cycles by using the fitting parameters as shown in Table 1.

| $Y$ | Temperature Range | Simulated Value | | |
|---|---|---|---|---|
| | | 1st Cycle | 500th Cycle | 1000th Cycle |
| Loss of $\Phi$ | R.T.–120 °C | 0.39 | 2.49 | 3.06 |
| | R.T.–140 °C | 0.57 | 2.58 | 3.15 |
| | R.T.–160 °C | 14.86 | 28.15 | 30.22 |
| | R.T.–180 °C | 24.34 | 37.77 | 39.67 |
| Loss of $B$ | R.T.–120 °C | 0.77 | 8.58 | 11.23 |
| | R.T.–140 °C | 0.84 | 32.19 | 47.63 |
| | R.T.–160 °C | 34.29 | 50.19 | 52.27 |
| | R.T.–180 °C | 39.30 | 53.39 | 55.25 |

### 3.2. The Effect of Temperature Cycling on Microstructure

EDS analysis is carried out to confirm the elemental distributions in the original samples and the sample in the range R.T.—180 °C after the 125th cycle, as shown in Figure 4. It is clearly seen that a Zn coating layer on the surface of the original NdFeB magnet is relatively complete, and Nd and Pr elements enrich at the grain boundary with a slight oxidation. After temperature cycling, the Zn coating layer is strongly oxidized and fractured. The oxidation of the Zn coating layer should be attributed to the main causes for its fracture. Meanwhile, a serious oxidation of the Nd, Pr-rich phases is confirmed. The EDS line scan results (Figure 5) also show that oxidation caused by temperature cycling mainly occurs in the Zn coating layer and the Nd, Pr-rich phase at the grain boundaries, and that the oxygen contents gradually decreases from the outer layer to the inner layer. Neither IOZ nor $\alpha$-Fe precipitates are observed in our study, but which have been found in both cases of the bare NdFeB magnet and the NiCuZn-coating NdFeB magnet heat-treated at $\geq 700$ °C [16,17]. Furthermore, the fact that increase of the upper limit temperature does not cause the decrease of $H_{cj}$ (Figure 3c) also verifies that there are not soft magnetic $\alpha$-Fe precipitates in the cycled sample. These results show that the oxidation behavior of the NdFeB magnet under the temperature cycling with the upper limit temperature at $\leq 180$ °C is greatly different from that heat-treated at $\geq 700$ °C.

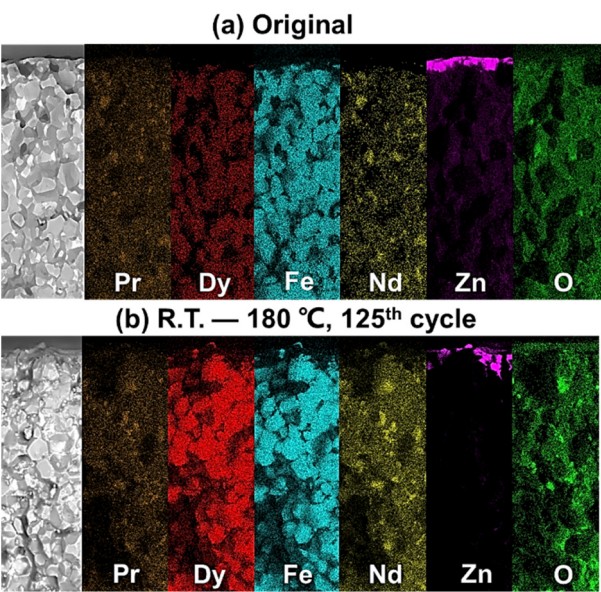

**Figure 4.** Cross-sectional backscattered SEM micrographs and EDS mappings of (**a**) the original sample and (**b**) the sample after 125th cycle from R.T. to 180 °C.

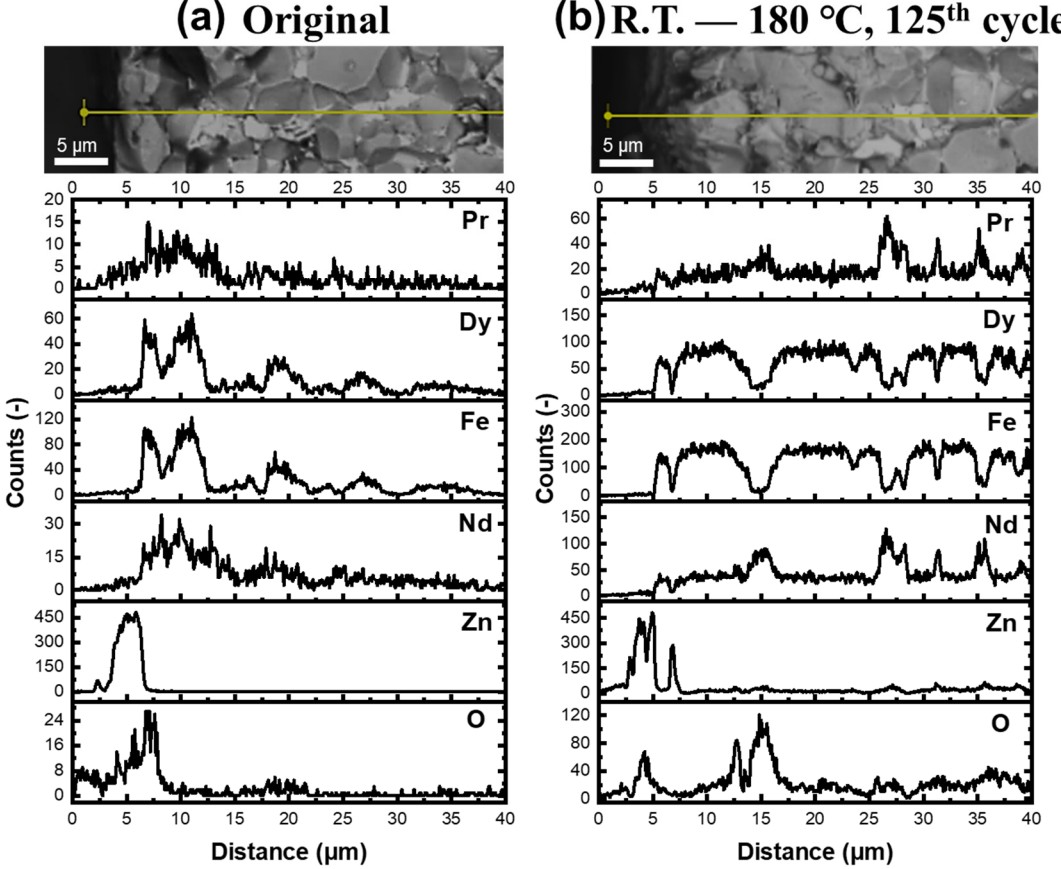

**Figure 5.** Cross-sectional backscattered SEM micrographs and EDS line scanning results of (**a**) the original sample and (**b**) the sample after 125th cycle from R.T. to 180 °C.

Compared with the original sample, the oxygen contents of the NdFeB grains existed on the outer surface of the cycled sample increase from 3.22 at.% to 8.76 at.%, and that existed on the inner surface increase from 1.77 at.% to 3.22 at.% (Figure S3). The change trend of the oxygen contents of Nd, Pr-rich phases is similar with that. It is believed that

the oxidation and fracture of the Zn coating layer during temperature cycling leads to the entry of O elements into the NdFeB matrix, and the diffusion channel of oxygen element is the Nd-, Pr-rich phase at the grain boundary. In our case, the oxidation of NdFeB grains should occur only on the grain surface rather than the overall decomposition of grains, due to the relatively lower upper limit temperature.

## 4. Conclusions

Losses of $\Phi$ and $B$ of the Zn-coating NdFeB magnets after temperature cycling are measured at different temperature ranges and can be well fitted by an index model. The loss difference of $\Phi$ and that of $B$ between the ranges of $-50$–$180\ ^\circ$C and R.T.–$180\ ^\circ$C were only 0.46% and $-0.29\%$ after 125 cycles, respectively (Table S2), demonstrating that the influence of lower limit temperature change on the magnetic degradation is limited. The losses of $\Phi$ and $B$ gradually increases at the upper limit temperature $\leq 140\ ^\circ$C but are mainly determined by the 1st cycle at $\geq 160\ ^\circ$C. Compared with $\Phi$, $B$ is more suitable as a leading indicator for judging the magnetic demand. At the upper limit temperature is $\leq 180\ ^\circ$C, and the temperature cycling plays a major role in the decreases of $B_r$ rather than $H_{cj}$. The oxidation the Zn coating layer during temperature cycling causes its fracture, which leads to the insertion of oxygen elements into the NdFeB matrix. The Nd-rich phases at the grain boundaries seem to provide a diffusion channel for oxygen elements, leading to the surface oxidation of NdFeB grains.

**Supplementary Materials:** The following supporting information can be downloaded at: https://www.mdpi.com/article/10.3390/coatings12050660/s1. Table S1: Temperature dependence of the $\Phi$, $B$ and their losses of the sample from R.T. to $300\ ^\circ$C; Table S2: The $\Phi$, $B$ and their losses of samples were cycled different times in the range of R.T.—$180\ ^\circ$C and $-50$–$180\ ^\circ$C, respectively; Table S3: Values of fitting parameter and their standard errors (*S.E.*) in the range of R.T.—$180\ ^\circ$C and $-50$–$180\ ^\circ$C, respectively. The fitting formulae is given by $Y = A \times (X - X_c)^P$; Table S4: The $\Phi$, $B$ and their losses of the samples were cycled different times in the range of R.T. to 120, 140, and $160\ ^\circ$C, respectively; Figure S1: (a) $\Phi$ and (b) its loss, (c) $B$ and (d) its loss of the samples cycled in ranges of R.T.—$180\ ^\circ$C and $-50$–$180\ ^\circ$C, respectively. The straight lines in (a) and (c) connect the data points. The lines in (b) and (d) are the fitting curves, and the fitting formula is given by $Y = A \times (X - X_c)^P$; Figure S2: The (a) $\Phi$ and (b) $B$ of the samples cycled in ranges of R.T. to 120, 140, 160, and $180\ ^\circ$C, respectively; Figure S3: Cross-sectional backscattered SEM micrographs and EDS point elemental analysis from the surface to the inner of the different NdFeB magnets. (a) original sample; (b) sample after 125 cycles in range of R.T.–$180\ ^\circ$C.

**Author Contributions:** Conceptualization and methodology, B.S. and X.-W.L.; validation, H.-L.W., Z.-M.L., and D.-L.C.; resources and equipment, Y.-L.M.; data curation, C.-Q.L. and K.-Q.S.; writing—preparation of the original, H.-L.W. and Z.-M.L. writing-review and editing, Z.-M.L.; funding acquisition, H.-L.W. and X.-W.L. All authors have read and agreed to the published version of the manuscript.

**Funding:** This study is part of the "Study on the environmental adaptability of typical magnetic materials and components" project, which is funded by the Development Fund of the Fifth Ninth Research Institute of China Ordnance Industry, grant number HDHDW5902020302.

**Institutional Review Board Statement:** Not applicable.

**Informed Consent Statement:** Not applicable.

**Data Availability Statement:** The data used to support the findings of this study are available from the corresponding author upon request.

**Conflicts of Interest:** The authors declare no conflict of interest.

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
