# Peer review of "The Effect of Temperature Cycling on the Magnetic Degradation and Microstructure of a Zn-Coated NdFeB Magnet"

_coatings, doi:10.3390/coatings12050660_

Round 1

Reviewer 1 Report

In this article the authors present a study on the magnetic property decay law of the NdFeB magnet with a Zn coating, at different temperatures, by using a temperature cycling experiment, to simulate the start–stop modes of electric motors. Moreover, a microstructural analysis ia also performed to evaluate the causes of the magnetic properties decay.

In general, the article is well written, and the research design is correct, the topic is interesting and applicative. The introduction is concise but concrete, it perfectly frames the problem and introduces the object of the research well. Weakness of the research is the lack of comparison with the uncoated equivalent. I suggest some improvements:

  1. A comparison with uncoated NdFeB magnet should be added.
  2. Table 1 should be better explained. It is not common to read 0.00±18.18 and found in the comment at line 129 “ …. Xc is negligible.”
  3. Figure 3: the error bars are missing
  4. 16: the authors are missing

Author Response

Please see the attachment. We put manuscripts, supplementary and responses in a PDF,  thanks.

Reviewer 2 Report

referee report 
coatings-1649016-peer-review-v1
Effect of Temperature Cycling on the Magnetic Properties and Microstructure of NdFeB Magnets
Hulin Wu et al.

The present manuscript discusses temperature cycling experiments of Nd-Fe-B magnets to simulate the working mode in a motor.
This is a very important issue for the application of these materials, thus this experiments is very interesting to a broad
readership. The topic is well suited for Coatings.
The present manuscript comprises 5 figures, 2 tables, and 21 references are given. A Supplemental Material is provided with
extra 2 figures and 4 tables.
The manuscript is well arranged, and the figures provided are well prepared -- only some figures are too small in the 
manuscript, which could be easily changed as there is no page limit. The English requires some improvement, so it is advisable
to have a native speaker checking the manuscript.

However, there are several points, both scientifically and technically which cause big troubles before a publication:
-- Please use also italics for the physical quantities throughout the text (lines 125-136). Also make sure that you use
     proper subscripts (e.g., B_r).
-- Dimensions of volumes should have proper units like mm^3.
-- Section 2.1: There should be a proper characterization of the materials used. There is no information about the magnetic
    specifications (surface field, force), nor an information concerning the Zn coating (how applied, how thick?). Even worse,
    somewhat later we learn that both new and recycled magnets were studied. This all must be properly described.
-- Section 2.2: You are not eliminating "magnetic effect" (for that, the samples must be demagnetized), but you minimize the
    magnetic interaction. Why the presence of Al is not checked by EDS? 
-- Figure 3: In Fig. 3b, data for coercivity and B_r are presented -- but it was not introduced in Section 2.1. as it should
    have been done. Be also aware that the audience of Coatings is not familiar with magnetic measurements, so please define
    the quantities properly.
    Figure 3a: What is the reason for labelling the axes in this funny way?
-- All figure captions should be reworked to properly describe the content of the figure.
-- Why there are ony EDS measurements for an original sample and a sample after 125 cycles?
-- Figure 4: Scale bar is too small.
-- In general, it would be useful to analyze the identical sample section before and after the treatments. This could be achieved
    by creating markers (e.g., mechanically or by FIB so that the SEM operator can re-locate the section). It is obvious from
    the images presented that the given sample sections have different chemical compositions.
-- As the manuscript is submitted to Coatings, I would expect a discussion of the effect of the Zn-coating. Does it change the
    properties measured?
-- The reference list is a catastrophe: The journal style is not followed, the references are given in different fashions.
    No proper journal abbreviations. Several references lack page or volume number, the papers of conferences are not properly
    cited. Also, chemical expressions in the titles must be formatted as well -- there is no Ndfeb!
-- The supplemental material is not given at the end of the manuscript.
-- What does mean "Date sharing is not applicable"? Yo should not share dates, but data. And the present article reports data!   

In summary, the present manuscript is not suitable for publication due to severe deficits.

Author Response

(The authors gave the same response as above.)

Reviewer 3 Report

The temperature cycling experiment was designed herein to simulate the working mode of the continuous start–stop operation of a motor and study its effects on the magnetic properties of NdFeB (28 × 24 × 4.5 mm) magnets with Zn coatings. By changing the temperature limits, the decay curves of the magnetic flux and the magnetic field strength for 125 cycles were recorded and found that the increased upper limit temperature was the main reason for performance degradation, while changes in the lower limit temperature had an almost negligible effect. The demagnetization curves after cycling verified the temperature cycle as the main cause of the reduced remanence and had little effect on the coercivity. Microstructure and element distribution analysis of the magnetic surface done by FE-SEM and EDS spectroscopy, revealed that the coating and matrix oxidation from the surface induced magnetic performance degradation. 
Manuscript can be accepted after minor revision with the following comments. 
1.    Date Availability Statement should be”Data Availability Statement”.
2.    Figure S2: present the Scale bar in the SEM images.
3.    Re. 16 Authors names are missing. 

Author Response

(The authors gave the same response as above.)

Reviewer 4 Report

The paper is well written, and the language clearly demonstrates technical competency. The figures and diagrams along with their explanation are very clear. In fact, it has been difficult to find discrepancies. But the authors have made this paper very brief, and this is in itself an issue because it omits some of the important details.
Hence, the authors need to address some minor issues

  1. In the introductory sections, the authors have missed a great deal of literature, the work of peers. How does the cycling affect the magnetic properties and what did the researchers observe?
  2. The introduction is weak because the novelty is not well highlighted, and it is not driven from systematic studies carried out so far why is it that this study clearly differentiates from others. Upon literature search, we can easily find many papers that are carrying out research in these or some of these underlying aspects that overlap this work.
  3. Section 3 is Results and discussion but there is no discussion compared with what has already been investigated by the peers.
  4. Section 3.2 does not discuss the issue in light of the literature review. Certainly, this issue would have been experienced by various other researchers
  5. Hence, the same story continues in the results as well, that the authors have not compared their results with other researchers. Although their magnitude might be different perhaps the trend could match.

In summary, please corroborate your work with the peer work and focus on strengthening the discussion.

Author Response

(The authors gave the same response as above.)

Round 2

Reviewer 2 Report

Well performed revision, all issues were addressed properly. Thus, the manuscript is now suitable for publication